# Relationship between Volpara Density Grade and Compressed Breast Thickness in Japanese Patients with Breast Cancer

**DOI:** 10.3390/diagnostics14151651

**Published:** 2024-07-31

**Authors:** Mio Adachi, Toshiyuki Ishiba, Sakiko Maruya, Kumiko Hayashi, Yuichi Kumaki, Goshi Oda, Tomoyuki Aruga

**Affiliations:** 1Department of Surgery (Breast), Tokyo Metropolitan Cancer and Infectious Disease Center Komagome Hospital, Tokyo 113-8677, Japan; adcsrg2@tmd.ac.jp (M.A.); tomoyuki_aruga@tmhp.jp (T.A.); 2Department of Breast Surgery, Tokyo Medical and Dental University, Tokyo 113-8510, Japan; maruya.sakiko@tmd.ac.jp (S.M.); hayashi.srg2@tmd.ac.jp (K.H.); kumaki.srg2@tmd.ac.jp (Y.K.); odasrg2@tmd.ac.jp (G.O.)

**Keywords:** breast density, dense breast, Volpara, compressed breast thickness, pseudo-dense breast

## Abstract

Background: High breast density found using mammographs (MGs) reduces positivity rates and is considered a risk factor for breast cancer. Research on the relationship between Volpara density grade (VDG) and compressed breast thickness (CBT) in the Japanese population is still lacking. Moreover, little attention has been paid to pseudo-dense breasts with CBT < 30 mm among high-density breasts. We investigated VDG, CBT, and apparent high breast density in patients with breast cancer. Methods: Women who underwent MG and breast cancer surgery at our institution were included. VDG and CBT were measured. VDG was divided into a non-dense group (NDG) and a dense group (DG). Results: This study included 419 patients. VDG was negatively correlated with CBT. The DG included younger patients with lower body mass index (BMI) and thinner CBT. In the DG, patients with CBT < 30 mm had lower BMI and higher VDG; however, no significant difference was noted in the positivity rate of the two groups. Conclusions: Younger women tend to have higher breast density, resulting in thinner CBT, which may pose challenges in detecting breast cancer on MGs. However, there was no significant difference in the breast cancer detection rate between CBT < 30 mm and CBT ≥ 30 mm.

## 1. Introduction

There are an estimated 220 million patients with breast cancer worldwide [1,2]. In 2020, 68,000 people died from breast cancer, making it the fifth leading cause of death globally [2,3]. Breast cancer is the most common cancer among Japanese women, prompting health and welfare efforts to reduce the associated mortality rate. In 2004, the Ministry of Health, Labour and Welfare initiated programs to conduct mammographs (MGs) for breast cancer for women aged >40 years [4]. The USPSTF concludes with moderate certainty that biennial screening mammography in women aged 40 to 74 years has a moderate net benefit [5]. However, in addition to the risk of radiation exposure and discomfort, MG screening presents other challenges, such as false negatives. This indicates that despite undergoing MG screening, cases of early-stage breast cancer might not be detected, especially among patients with dense breasts and high breast density [6,7]. High breast density on MGs indicates an increased risk of breast cancer, with a 3–6 times higher risk for patients with denser breasts than for those with fatty breasts. Thus, effective communication and raising awareness regarding this risk are essential [6,7,8,9]. Breast density is influenced by various factors, including age, weight, body mass index (BMI), family history, genetic predisposition, and parity [10]. Japanese women have a higher breast density than Western women [5,11,12]. It is known that adding ultrasound (US) to MG increases sensitivity; however, US has problems with a high false-positive rate and the fact that diagnostic accuracy depends on the skill of the operator [12]. Dynamic contrast-enhanced (DCE) magnetic resonance imaging (MRI) has very high sensitivity but low specificity [13]. In addition, repeated use of gadolinium contrast media causes side effects, increases costs, and deposits gadolinium in the body [14]. DCE-MRI screening is useful for high-risk breast cancer patients but is considered over-testing in non-high-risk patients [13]. Although 18F-fluorodeoxyglucose (FDG)/positron emission tomography (PET) with computed tomography (CT) is excellent for staging advanced breast cancer, it is not suitable for screening due to its limited detection of tumors with low metabolic activity and small breast cancers [15,16].

The 5th edition of the Breast Imaging Reporting and Data System now defines breast density, unlike previous versions [16]. However, initial evaluations of mammary gland density can be subjective and are not quantifiable [17,18]. Therefore, tools such as QUANTRA and Volpara were developed to measure mammary gland density; these have been approved by the Food and Drug Administration (FDA) [19,20]. Volpara (Matakina Technology Limited, Wellington, New Zealand) calculates the mean volume percentage of breast volume (Volpara density grade [VDG]%) using digital mammograms [20,21]. VDG represents the volume of fibroglandular tissue divided by the total volume of the breast. However, such breast density measurement software is only available in a limited number of medical institutions, making it difficult to objectively measure breast density for all patients. Therefore, the correlation between indices that can be measured at any facility during MG and mammary gland density should be explored to establish a standard for assessing breast density in facilities lacking the software to measure mammary gland t density [22]. Moshina et al. reported that mammary glands with low compressed breast thickness (CBT) have high breast density and reported a negative correlation between CBT and VDG [23]. Therefore, we hypothesized that by examining the correlation between CBT and VDG, CBT could serve as an objective indicator of mammary gland density. However, no studies have clarified the relationship between breast density, CBT, age, weight, and BMI among Japanese women; moreover, no study has examined the correlation between VDG and CBT in women diagnosed with breast cancer. Breasts with low CBT tend to have high breast densities [22]. Noma et al. reported that CBT < 30 mm does not increase the risk of false-negative MG results despite high breast density; they classified such breasts as pseudo-dense breasts [24]. The Japan Breast Cancer Screening System Management Organization, a Japanese nonprofit organization, assesses breast density based on reports indicating that the detection rate of breast cancer is not lower for dense breasts with CBT < 30 mm compared with non-dense breasts. They recommend classifying CBT < 30 mm as indicative of being non-dense when physicians are uncertain about categorizing breast density. It defines CBT < 30 mm as a pseudo-dense breast. However, reports in English on dense breasts with a CBT < 30 mm are scarce. The Asian population has a higher breast density than the Western population [5,11]. Therefore, in this study, we aimed to elucidate the relationship between breast density and CBT among Japanese women with breast cancer.

## 2. Materials and Methods

### 2.1. Patients

Women who underwent MG before breast cancer surgery at Tokyo Metropolitan Komagome Hospital between February 2019 and April 2020 were included in this study. Patients with a history of breast surgery, synchronous or metachronous bilateral breast cancer, and de novo stage IV breast cancer were excluded.

### 2.2. Data Collection

MG was performed in two directions: craniocaudal (CC) and mediolateral oblique (MLO). Digital MG systems (GE Essential, Siemens Novation, Hologic Selenia) were used for imaging, and five radiology technicians conducted the examinations. CBT is automatically measured during MG screening, recorded as four values: left, right, MLO, and CC [21]. MG density and CBT measurements were taken from the nonmalignant side of the breast. In some cases, only the MLO measurement is performed in Japanese resident screening. Therefore, we used the MLO value as the measurement index in this study. Diagnoses were made by two certified MG interpreters following the Japanese image interpretation guidelines. Category (C-)1 indicates no findings; C-2 indicates benign findings; and C-3, C-4, and C-5 require further investigation. Patients with C-3 or above were defined as those in whom the lesion could be identified via MG [22]. Mammary gland density was evaluated using Volpara (VolparaTM, Matakina Technology Limited, New Zealand). Volpara assesses density based on VDG, categorized as follows: (a) fatty (Volpara < 3.5%), (b) scattered (Volpara 3.5–7.5%), (c) heterogeneous (Volpara 7.5–15.5%), and (d) extremely high concentration (Volpara > 15.5%). The patients were categorized into two groups: the non-dense group (NDG, VDG a and b) and the dense group (DG, VDG c and d) [16]. T classification was determined on postoperative pathology for patients without preoperative drug therapy, and the larger tumor diameter was observed on ultrasound (US) or magnetic resonance imaging for patients undergoing preoperative drug therapy. Axillary lymph node metastasis in patients undergoing preoperative drug therapy was assessed based on pathological examination, including cytological and histological examinations performed before treatment. When pretreatment pathological examination was not possible, axillary lymph node dissection was conducted after imaging studies suggesting lymph node metastasis, whereas patients considered negative underwent sentinel lymph node biopsy. A pathological diagnosis was made using postoperative specimens for patients not undergoing preoperative drug therapy and histological diagnosis before treatment for those undergoing preoperative drug therapy. Similarly, the pathological stage was determined based on surgical specimens for patients not undergoing preoperative drug therapy and preoperative imaging tests for those undergoing preoperative drug therapy.

### 2.3. Statistical Analysis According to Prognosis

Clinical information was retrospectively collected by examining medical records, and statistical analysis was performed using EZR software (EZR version 1.61) [23].

### 2.4. Ethical Approval and Consent for Participation

This study protocol adhered to the principles of the Declaration of Helsinki, the Clinical Research Act (Act No. 16 of 2017), the Enforcement Regulations of the Clinical Research Act (Ministry of Health, Labor and Welfare Ordinance No. 17 of 2018), and related notices. This study was conducted using data approved by the Ethics Review Committee of our hospital. Patients were informed that their clinical data would be used for research purposes, and comprehensive consent was obtained for this study.

## 3. Results

A total of 419 patients were included in this study, with a median age of 54 years (range: 31–93), median height of 156.1 cm (range: 132.5–174.0 cm), and median weight of 53.2 kg (range: 35.3–97.3 kg) (Table 1). By age group, 124 (30%) patients were aged <40 years, 27 (6%) were aged 40–49 years, 106 (25%) were aged 50–59 years, 60 (14%) were aged 60–69 years, 99 (24%) were aged 70–79 years, and 33 (8%) were aged ≥80 years (Figure 1). The patients’ median BMI was 22.2 (range: 15.3–39.8). BMI was categorized into three levels: ≤18.5, thin; 18.5–25, normal body type; and ≥25, obesity index, according to the general standard. Thirty-seven (8%) patients had a BMI of <18.5, and ninety-four (22%) had a BMI of ≥25 (Figure 2). The reason for their visit was indicated as follows: 211 (51%) had only subjective symptoms, 180 (43%) showed abnormalities in screening without symptoms, 16 (3%) had abnormalities in screening with symptoms, and 11 (3%) were undergoing follow-up for other diseases. Moreover, 282 (67%) masses were found to be palpable, 105 (25%) were not palpable, and 32 (8%) were unknown (Table 1).

We also investigated clinical and pathological factors in these patients. The T classification was Tis in 52 (12%) patients, T1 in 245 patients (58%), T2 in 109 patients (26%), T3 in 10 patients (2%), and T4 in 4 patients (1%). The pathological stage was stage 0 in 52 (12%) patients, stage I in 211 (50%), stage II in 134 (33%), and stage III in 22 (5%).

The pathologies included 52 (12%) patients with ductal carcinoma in situ (DCIS), 344 (82%) with invasive ductal carcinoma, and 23 (5%) with other invasive cancers (Table 2).

Next, we examined the findings of MG. A total of 49 (12%) patients were classified as C-1 and C-2, 97 (23%) as C-3, 189 (45%) as C-4, and 84 (20%) as C-5. MG revealed masses in 112 (27%) patients, calcification in 69 (17%), asymmetric density in 40 (10%), and architectural distortion (with overlap) in 39 (10%). VDG was a (fatty) in 7 (3%) patients, b (scattered) in 64 (15%), c (heterogeneous high concentration) in 164 (39%), and d (extremely high concentration) in 184 (43%). There were 71 (18%) patients with a and b combined in the NDG and 348 (82%) with c and d combined in the DG. Median CBT was 40.95 mm (range 11.7–81.8 mm) (Table 3).

We then categorized CBT into five groups (≤20, 20–29, 30–39, 40–49, and ≥50 mm) and examined its relationship with VDG. Among patients with CBT ≥50 mm, approximately half were classified into the DG, whereas among those with CBT < 30 mm, all were classified into the DG. A trend of higher VDG with smaller CBT values was noted, indicating that breasts with smaller CBT tended to have higher breast density (*p* < 0.01) (Figure 3 and Figure 4c).

We investigated the relationship of VDG with BMI, age, and CBT. We observed that lower BMI was associated with higher VDG, lower age was associated with higher VDG, and lower CBT values were associated with higher VDG (Figure 4). In other words, a tendency toward higher breast density was observed in individuals with lower BMI, younger age, and lower CBT values. We then categorized VDG into a DG and an NDG (Table 3). The NDG comprised 71 (18%) patients, and the DG comprised 348 (82%). The median age ± standard deviation (SD) of NDG patients was 64.5 ± 11.0 years, whereas that of DG patients was 55.8 ± 11.0 years, with NDG patients tending to be older than DG patients (*p* < 0.01). The mean CBT ± SD of NDG patients was 54.5 ± 10.9 mm, whereas that of DG patients was 39.0 ±11.6 mm, with NDG patients tending to have larger CBT (*p* < 0.01). The mean BMI ± SD of NDG patients was 25.9 ± 4.6, whereas that of DG patients was 22.1 ± 3.2, with NDG patients tending to have higher BMI (*p* < 0.01). For patients with MG of C-3, C-4, and C-5, there was a tendency for a higher rate of lesion identification on MGs in NDG patients (68 [96%]) compared with DG patients (302 [87%]) (*p* = 0.02) (Table 4). Although DG patients are generally believed to have less fat than NDG patients, there have been reports of apparently dense breasts with CBT < 30 mm, which does not decrease the breast cancer detection rate [20]. Therefore, we further divided the DG into two subgroups: patients with CBT ≥ 30 mm and those with CBT < 30 mm (Table 5). The mean age ± SD was 56.8 ± 16.0 years for the CBT < 30 mm group and 55.4 ± 12.9 years for the CBT ≥ 30 mm group; no significant difference was observed between the two groups (*p* = 0.39). Additionally, the mean BMI ± SD was 19.7 ± 2.1 for the CBT < 30 mm group and 22.9 ± 3.1 for the CBT ≥ 30 mm group; patients in the CBT < 30 mm group tended to be underweight (*p* < 0.01). The mean VDG ± SD was 19.8% ± 6.0% for the CBT < 30 mm group and 15.7% ± 6.3% for the CBT ≥ 30 mm group; breast density tended to be higher in the CBT < 30 mm group (*p* < 0.01). The number of patients in which C-3 or higher was detected through MGs was 78 (91%) in the CBT < 30 mm group and 224 (85%) in the CBT ≥ 30 mm group; no significant difference was observed between the two groups (*p* = 0.62). In the DG, the CBT < 30 mm group had older patients, thinner CBT, and higher mammary gland density compared with the CBT ≥ 30 mm group. Although not statistically significant, the positive rate of breast cancer tended to be higher in the CBT < 30 mm group.

## 4. Discussion

In this study, among 419 patients with breast cancer, 344 (82%) were found to have dense breasts using VDG and a median CBT of 40.4 mm. Higher breast density was correlated with younger age, lower BMI, and smaller CBT. It was also more difficult to identify breast cancer in MG in the DG than in the NDG. In the DG, patients in the CBT < 30 mm group tended to have lower BMI and higher breast density than those in the CBT ≥ 30 mm group. Although there was no significant difference in the proportion of C-3, C-4, and C-5 patients between the CBT < 30 mm and CBT ≥ 30 mm groups, the proportion tended to be higher for the CBT < 30 mm group. These findings align with those reported by Mehnati et al., who reported higher breast density, younger age, lower BMI, and smaller CBT [25]. Similar results were reported by Mohina et al., with CBT tending to decrease with increasing mammary gland density [23]. When breast density was categorized into four levels and compared with CBT, higher breast density corresponded to thinner CBT. Furthermore, DG patients had a lower likelihood of breast cancer detection on MGs compared with NDG patients.

In this study, the median CBT was 40.4 mm. Du et al. reported a CBT of 42 mm in a Chinese patient [26], which closely aligns with the value observed in the present study’s Japanese patients. It is suggested that CBT may be smaller in Asians than in Europeans [27]. Lau et al. reported a median CBT of 53.17 ± 12.18 mm and a median VDG of 9.76% ± 5.82% in Asian women [28].

J-START is contemplating the inclusion of US alongside MG for women in their 40s. Research has shown that supplementing MG with US for breast cancer screening not only enhances the sensitivity of cancer detection but also reduces the occurrence of interval breast cancer [29]. Considering these findings, additional testing alongside US could prove beneficial for younger individuals, as breast cancer detection via MG alone is difficult due to their high breast density. In our study, DG patients were younger than NDG patients. Moreover, it has been proposed that dense breasts with small CBT may represent apparent dense breasts. To investigate whether these are true dense breasts or pseudo-dense breasts, we subcategorized DG patients based on CBT < 30 mm. Although the number of patients with MG classifications C-3, C-4, and C-5 did not significantly differ between the CBT < 30 mm and CBT ≥ 30 mm groups, there was a tendency for a higher proportion of patients with MG classifications C-3, C-4, and C-5 in the CBT < 30 mm group. However, the distinction between pseudo-dense and true dense breasts and their clinical significance remains uncertain, warranting further investigation.

This study has some limitations. First, only patients with breast cancer were included; healthy individuals were not considered. Previous reports indicate that patients with breast cancer often exhibit high breast density [6], and the proportion of dense breasts in this study may be higher than normal [30]. Second, although DG and NDG patients were differentiated based on VDG, DG patients accounted for two-thirds of the study cohort. Furthermore, since this study included patients with subjective symptoms, it may not entirely represent true screening. Although there is hope for the addition of US to breast cancer screening for highly dense breasts in Japan, it is yet to commence. This study suggests that younger age, lower BMI, and smaller CBT correlate with higher chances of dense breasts. Currently, breast density is subjectively assessed by image readers, leading to varying decisions on density notification across local governments. However, with CBT, existing mammograms can be assessed, potentially aiding in determining breast density. Furthermore, among patients with breast cancer, the proportion of MG classifications C-3, C-4, and C-5 was low in breasts with high mammary gland density. Although there are challenges in introducing US in local government examinations due to time and personnel constraints, CBT can assist in determining the need for US. For instance, if the CBT is <30 mm, the possibility of high breast density is low, so omitting US could be considered if the examinee does not request it. In the future, it is hoped that objective breast density will be indicated at every screening and fed back to the examinee. Until then, however, judgment of breast density will be subjective. CBT will be one of the factors used to judge breast density. When subjectively judging a dense breast, or when determining whether it is a high-density breast if the CBT is <30 mm, the alert for a high-density breast may be a weak recommendation compared to those who have a CBT of ≥30 mm.

## 5. Conclusions

This study revealed that younger women tend to have higher breast density, and as breast density increases, CBT tends to decrease, posing difficulties in breast cancer detection via MG. Notably, 14% of dense breasts were MG-negative, underscoring the need to explore alternative breast cancer screening modalities.

## Figures and Tables

**Figure 1 diagnostics-14-01651-f001:**
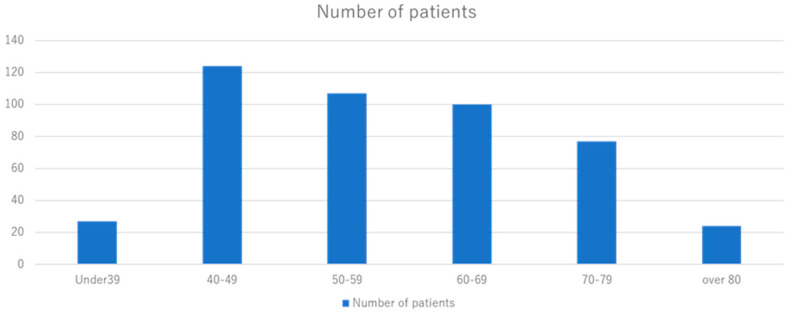
Number of patients by age group. A total of 124 (30%) patients were aged <40 years, 27 (6%) were aged 40–49 years, 106 (25%) were aged 50–59 years, 60 (14%) were aged 60–69 years, 99 (24%) were aged 70–79 years, and 33 (8%) were aged ≥80 years.

**Figure 2 diagnostics-14-01651-f002:**
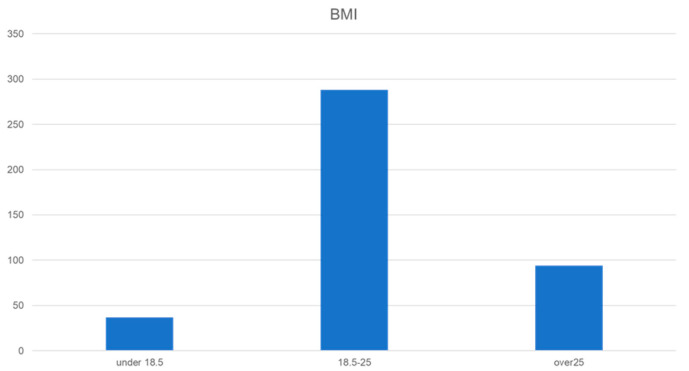
BMI of the patients. Thirty-seven people (8%) had a BMI <18.5, and ninety-four (22%) had a BMI ≥25.

**Figure 3 diagnostics-14-01651-f003:**
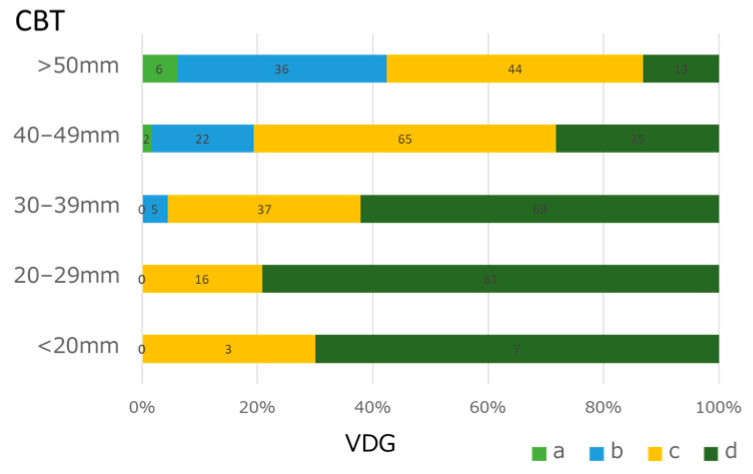
VDG based on CBT classification The thinner the CBT, the higher the percentage of hyperintense breasts tends.

**Figure 4 diagnostics-14-01651-f004:**
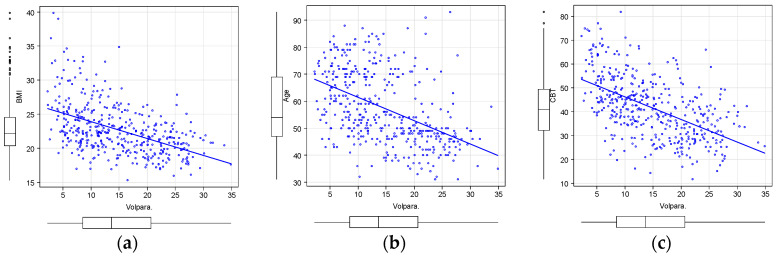
Relationship between VDG, BMI, age, and CBT. Each individual is shown in the plot. Straight lines indicate regression curves. (**a**) Relationship between VDG and BMI. Correlation coefficient is −0.543. *p* < 0.01. (**b**) Relationship between VDG and age. Correlation coefficient is −0.475. *p* < 0.01. (**c**) Relationship between VDG and CBT. Correlation coefficient is −0.485. *p* < 0.01. There is a negative correlation between VDG and BMI, age, and CBT.

**Table 1 diagnostics-14-01651-t001:** Patient characteristics.

	n = 419	%
Mean age, years (range)	54 (31–93)	
Mean height, cm (range)	156.1 (132.5–174.0)	
Mean body weight, kg (range)	53.2 (35.3–97.3)	
Mean BMI (range)	22.2 (15.3–39.8)	
Chief compliant		
Subjective symptoms	211	51%
Abnormal health check-up findings without subjective symptoms	180	43%
Abnormal health check-up findings with subjective symptoms	16	3%
During the follow-up of other diseases	11	3%
Unknown	1	0.2%

**Table 2 diagnostics-14-01651-t002:** Pathological features.

Clinical T	n	%
Tis	52	12%
T1	244	58%
T2	109	26%
T3	10	2%
T4	4	1%
Pathological stage		
Stage 0	52	12%
Stage I	211	50%
Stage II	134	33%
Stage III	22	5%
Pathology		
Ductal carcinoma in situ	52	12%
Invasive ductal carcinoma	344	82%
Other	23	5%
Subtype		
Luminal	335	80%
Luminal-HER2	25	6%
HER2	35	8%
Triple-negative	21	5%
Unknown	3	0.70%
Neoadjuvant therapy		
Yes	47	12%
No	372	88%

**Table 3 diagnostics-14-01651-t003:** MG findings.

MG Category		%
1 and 2	49	12%
3	97	23%
4	189	45%
5	84	20%
MG findings ※		
Mass	112	27%
Calcification	69	17%
Focal asymmetric density	40	10%
Architectural distortion	39	10%
VDG		
Non-dense group	75	18%
a: fatty breast (VDG: under 3.5%)	7	3%
b: fatty breast with scattered fibroglandular densities (VDG: between 3.5% and 7.5%)	64	15%
Dense group	348	82%
c: heterogeneously fibroglandular breast (VDG: between 7.5% and 15.5%)	164	39%
d: extremely fibroglandular parenchyma (VDG: over 15.5%)	184	43%
Median CBT, mm (range)	40.95 (11.7–81.8)	

MG, mammography; VDG, Volpara density grade; CBT, compressed breast thickness. ※ There are duplicate cases.

**Table 4 diagnostics-14-01651-t004:** Comparison of the non-dense and dense groups.

	Non-Dense Group (n = 71)	Dense Group (n = 348)	*p* Value
Mean age ± SD, years (range)	64.5 ± 11.0 (43–84)	55.8 ± 11.0 (31–93)	<0.01
Mean CBT ± SD (range)	54.5 ± 10.9 (33.1–77.0)	39.0 ± 11.6 (11.7–81.8)	<0.01
Mean BMI ± SD (range)	25.9 ± 4.6 (19.2–39.9)	22.1 ± 3.2 (15.3–34.8)	<0.01
MG ≥ C-3	68 (96%)	302 (87%)	0.02
Clinical T			Under T1 vs. over T2 0.67
Tis	7 (9%)	45 (13%)	
T1	45 (65%)	199 (57%)	
T2	18 (25%)	91 (26%)	
T3	0 (0%)	10 (3%)	
T4	1 (1%)	3 (0.8%)	

SD, standard deviation; CBT, compressed breast thickness; MG; mammography.

**Table 5 diagnostics-14-01651-t005:** Classification of dense breasts using a CBT cutoff value of 30 mm.

	CBT < 30 mm (n = 86)	CBT ≥ 30 mm (n = 262)	*p* Value
Mean age ± SD, years (range)	56.8 ± 16.0	55.4 ± 12.4	0.39
Mean BMI ± SD (range)	19.7 ± 2.1	22.9 ± 3.1	<0.01
Mean VDG ± SD, % (range)	19.8 ± 6.0	15.7 ± 6.3	<0.01
MG categories 3, 4, and 5	78 (91%)	224 (85%)	0.64

CBT, compressed breast thickness; SD, standard deviation; VDG, Volpara density grade; MG, mammography.

## Data Availability

Raw data were generated at Tokyo Metropolitan Cancer and Infectious Disease Center Komagome Hospital. Derived data supporting the findings of this study are available from the corresponding author, M.A., on request.

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
