# Peer review of "Relationship between Volpara Density Grade and Compressed Breast Thickness in Japanese Patients with Breast Cancer"

_diagnostics, 2024, doi:10.3390/diagnostics14151651_

Round 1
Reviewer 1 Report
Comments and Suggestions for Authors Thank you very much for this interesting manuscritpt. Discussion, lines 227-228: "Higher breast density was correlated with younger age, lower BMI, and smaller CBT. Additionally, DG patients tended to be younger, with lower CBT values and BMI compared with NDG patients." Can you write only one sentence? These two sentences seem to be almost identical. Comments on the Quality of English LanguageGood quality of English language
Only minor typo errors should be corrected
Introduction line 72: change "difines" into "defines"
Discussion, line 248: change "intermediate breast cancer" into "interval breast cancer"
Discussio, last sentence: I cannot perfectly understand what do you want to say, please think about modifiy this sentence: "For instance, if CBT ≥30 mm, or during patient consultations, CBT could guide deci- 276 sions regarding the recommendation level for additional US."
Reviewer 2 Report
Comments and Suggestions for Authors
1. What other methods are there for early detection of breast cancer?
2. I recommend that it should be stated in the text of the article that when is it better for women to start doing breast examinations and screening?
3. I offer some examples of similar studies for comparison. Choose a few of them (to your liking) for comparison and discuss them in the text of the article:
· Relationship between volumetric breast density estimated by VolparaTM and mammographic percent density by Cumulus. (Karahalios et al, 2020)
· Association Between Automatically Quantified Volumetric Breast Density and Breast Cancer Risk: A Nested Case–Control Study. (Niger et al, 2020)
· Association Between Breast Density and Compressed Breast Thickness in Korean Women Undergoing Full-Field Digital Mammography. (Cho et al, 2018)
· Volumetric breast density assessment: Comparison of volumetric breast density between white and Asian women. (Carney et al, 2016)
· Can Digital Breast Tomosynthesis Replace Digital Mammography as the Primary Screening Tool for Average-Risk Women? (Ko et al, 2018)
· Digital breast tomosynthesis versus digital mammography: comparison of multifocal cancer detection rates and interobserver agreement. (Valentin et al, 2016)
· Automated breast density evaluation: Reliability of Volpara, Quantra, and Cumulus software compared to visual assessment. (Keller et al, 2016)
4. Suggest some strategies for long-term follow-up of patients and evaluation of clinical results.
5. The titles of the Tables & Figures are not complete and sufficient information cannot be extracted from them.
6. In Figure 4, the curves and the correlation coefficient of the relationships are not shown.
Round 2
Reviewer 2 Report
Comments and Suggestions for Authors
Authors responded on all my queries in satisfactory way. I do not have more comments.